# Chronic Lymphocytic Leukemia-Induced Humoral Immunosuppression: A Systematic Review

**DOI:** 10.3390/cells9112398

**Published:** 2020-11-02

**Authors:** Ewelina Grywalska, Monika Zaborek, Jakub Łyczba, Rafał Hrynkiewicz, Dominika Bębnowska, Rafał Becht, Barbara Sosnowska-Pasiarska, Jolanta Smok-Kalwat, Marcin Pasiarski, Stanisław Góźdź, Jacek Roliński, Paulina Niedźwiedzka-Rystwej

**Affiliations:** 1Department of Clinical Immunology and Immunotherapy, Medical University of Lublin, 20-093 Lublin, Poland; ewelina.grywalska@gmail.com (E.G.); zaborelli00@gmail.com (M.Z.); jakublyczba7@gmail.com (J.Ł.); jacek.rolinski@gmail.com (J.R.); 2Institute of Biology, University of Szczecin, Felczaka 3c, 71-412 Szczecin, Poland; rafal.hrynkiewicz@gmail.com (R.H.); bebnowska.d@wp.pl (D.B.); 3Clinical Department of Oncology, Chemotherapy and Cancer Immunotherapy, Pomeranian Medical University of Szczecin, Unii Lubelskiej 1, 71-252 Szczecin, Poland; rbecht@pum.edu.pl; 4Department of Oncocardiology, Holy Cross Cancer Centre, 25-734 Kielce, Poland; spbasia@gmail.com; 5Department of Clinical Oncology, Holy Cross Cancer Centre, 25-734 Kielce, Poland; jolantasmok1@gmail.com (J.S.-K.); stanislawgozdz1@gmail.com (S.G.); 6Department of Immunology, Faculty of Health Sciences, Jan Kochanowski University, 25-317 Kielce, Poland; marcinpasiarski@gmail.com; 7Department of Hematology, Holy Cross Cancer Centre, 25-734 Kielce, Poland; 8Faculty of Medicine and Health Sciences, The Jan Kochanowski University, 25-516 Kielce, Poland

**Keywords:** chronic lymphocytic leukemia, CLL, immune system, humoral immunodeficiency, secondary immunodeficiency disorders, infection, targeted therapy

## Abstract

Secondary immunodeficiency is observed in all patients with chronic lymphocytic leukemia (CLL) in varying degrees. The aim of the study was to review the available literature data on patients with CLL, with particular regard to the pathogenesis of the disease and the impact of humoral immunity deficiency on the clinical and therapeutic approach. A systematic literature review was carried out by two independent authors who searched PubMed databases for studies published up to January 2020. Additionally, Google Scholar was used to evaluate search results and support manual research. The search resulted in 240 articles eligible for analysis. After all criteria and filters were applied, 22 studies were finally applied to the analysis. The data analysis showed that the clinical heterogeneity of CLL patients correlates with the diversity of molecular abnormalities determining the clinical picture of the disease, the analysis of which enables setting therapeutic targets. Additionally, in improving the therapeutic method, it is worth introducing supportive therapies with the use of vaccines, antibiotics and/or immunoglobins. Moreover, humoral immunodeficiency in CLL has a strong influence on the risk of infection in patients for whom infections are a major cause of morbidity and mortality.

## 1. Introduction

Secondary immunodeficiencies can be caused by numerous diseases and pharmacological therapies, and are becoming more common. Disease-related causes include lymphoproliferative neoplasms such as chronic lymphocytic leukemia (CLL) and multiple myeloma. All chronic lymphocytic leukemia patients are characterized by an immunodeficiency to some extent. Several pathways of the immune system are impaired by the CLL, including both immunodeficiency related to the leukemia itself (humoral and cellular immune dysfunction) and the results of cumulative immunosuppression resulting from CLL-specific treatment—chemoimmunotherapy [1,2]. According to the 2008 World Health Organization (WHO) classification, Chronic Lymphocytic Leukemia is defined as a lymphoproliferative disorder characterized by a progressive monoclonal absolute lymphocytosis of small, mature-looking CD5^+^/CD19^+^ B cells’ accumulation in the peripheral blood (PB), bone marrow (BM) and lymphatic organs [3]. Chronic lymphocytic leukemia is the most common adult leukemia in Europe and in North America. CLL is most frequently diagnosed among people aged 65–74. Median age at diagnosis is 70. Men are diagnosed almost twice as often as women (6.8 to 3.5 ratio). The rate of new cases is estimated at 5 per 100,000 population [4]. The clinical presentation of CLL is diverse, varying from an indolent, stable disease to those cases with aggressive leukemia who succumb to their disease, which ends in death in a short time. Most patients are asymptomatic at the moment of diagnosis and the disease is detected due to increased lymphocyte count at blood evaluations performed for unrelated reasons. In some patients, the disease proceeds slowly and, consequently, treatment is not required for many years. In such cases, a “wait and see” strategy seems justified. In other patients, the main clinical problem is complications of the underlying disease, including autoimmune hemolytic cytopenia, hipogammaglobulinemia and increased susceptibility to infections, and rarely hypersplenism. Despite advances in treatment and improved survival, infections account for 50–60% of all deaths, which is a major source of morbidity and mortality in patients with chronic lymphocytic leukemia [5].

Extended duration of the disease and immunosuppressive long-term therapy increase the risk of infectious complications. In early untreated stages of CLL, bacterial infections are mainly associated with hipogammaglobulinemia. However, in advanced stages, the researcher’s attention is drawn to the intake of purine analogues, neutropenia and impaired cellular immunity, increasing susceptibility to infections [6].

## 2. Chronic Lymphocytic Leukemia—Diagnosis, Clinical Features and Prognostic Factors

In recent years, many new factors have been established, and become the basis for the development of therapeutic strategies. These include immunophenotypic markers of leukemia cells (CD38 and ZAP-70) assessed by flow cytometry in the immunoglobulin variable region of the heavy chain gene (IgVH). In combination with other immunophenotypic parameters of leukemia cells, including CD5 and CD23, they also allow for a more detailed evaluation of the effectiveness of therapy and its further monitoring at the level of minimal residual disease (MRD). Although such a strategy can only be considered in a few patients, detailed molecular diagnostics and MRD monitoring is necessary to rationally select the available therapy methods in the optimal time of the entire treatment process [7].

The use of novel molecular biology techniques revealed significant genetic and epigenetic heterogeneity among patients with diagnosed CLL and helped to define novel somatic mutations of prognostic value. Functional B cell receptors (BCR) expressed on the surface of chronic lymphocytic leukemia cells and superficial IG expression are crucial for the survival and function of normal B cells and many lymphoproliferative disorders [8]. The B-cell receptor is a protein complex consisting of antigen-specific sIg and heterodimers Igα and Igβ (CD79A and CD79B). Antigen binding to sIg activates Src family kinases (LYN), which phosphorylate immunoreceptor tyrosine-based activation motifs (ITAM) of Igα and Igβ cytoplasmic peptide tails [9]. ITAM phosphorylation can dock the SYK kinase and launch SYK-dependent signal transduction—activation of Bruton’s kinase (Bruton’s tyrosine kinase (BTK)) and phosphatidylinositol-3-kinases (phosphoinositide 3-kinase (PI3K)). Next, activation of phospholipase C-γ2 (phospholipase C-γ2 (PLC-γ2)), protein kinase C signaling pathways (protein kinase C (PKC)), κB (nuclear factor kappa-light-chain-enhancer of activated B cells (NF-κB)) and Mitogen Activated Protein Kinases (MAPKs) occurs and a change in the concentration of calcium ions in the cell is observed. The ultimate consequence of antigen activation of the BCR is a change in the gene expression profile [9]. CLL has a distinct BCR signaling from normal B cells, which is characterized by a variable response to antigen stimulation, activation of anti-apoptotic signaling pathways and low IgM expression. Due to gene expression profiling, CLL cells share many features with antigen-activated mature B cells, suggesting a role in activating BCR signaling in the disease pathogenesis [10].

Despite the fact that CLL is a heterogeneous disease with a variable clinical course, the data suggest that the course of CLL disease can be divided by the presence or absence of somatic hypermutation of the immunoglobulin variable region of the heavy chain gene (IgVH), strongly associated with overexpression of ZAP-70 [10,11,12,13]. The ZAP-70 protein enhances the BCR signaling of blood-derived CLL cells through an adapter role independent of its kinase activity, and also enhances migration to chemokines and the response to survival stimuli from the microenvironment. A recent tissue comparison showed an enhanced upregulation of BCR-related genes by the microarray in the bone marrow and lymph nodes of all CLL patients compared to blood, regardless of ZAP-70 expression or immunoglobulin heavy chain variable region mutation status [13,14]. Thus, increased expression of BCR genes is present in all patients with CLL in lymph nodes and bone marrow. Moreover, patients with ZAP-70 positive disease have a higher BCR response [13,14]. Patients with CLL with mutated IGHV genes (IGHV-M CLL) have a better prognosis than patients with CLL with unmutated IGHV genes (IGHV-UM CLL). The most common genetic changes in CLL are deletions of 13q14 (del13q14) (50–60%). Most del13q14 deletions are monoallelic in nature and are more common in IGHV-M CLL than in IGHV-UM CLL [15,16,17].

The latest literature reports also emphasize the role of Del17p13; this alteration involves the tumor suppressor gene TP53, and, in more than 80% of cases, is associated with TP53 mutation on the remaining allele. This aberration is associated with a dire clinical outcome, being linked to tofludarabine-refractoriness, treatment resistance and early disease relapse [18]. In addition, cases that retain BCR signaling capability can be distinguished from those that do not by the presence of CD38 or ZAP-70 or the absence of an IgV mutation, all of the above being markers of poor prognosis, although these correlations are not absolute. This suggests that the ability of B-CLL cells to retain signaling capacity affects survival and clonal expansion, and negatively affects the patient’s clinical course [19].

Studies were also undertaken to establish if the relationship between monoclonal gammopathy and serum immunoglobulin levels may serve as a prognostic factor in CLL [20]. The study involved 1505 patients with CLL, divided on the basis of Ig aberrations at diagnosis, IgM monoclonal gammopathy (IgM/CLL), IgG monoclonal gammopathy (IgG/CLL), hipogammalobulinemia and normal Ig levels, respectively [20]. It was concluded that patients with any type of Ig abnormality were characterized by a shorter treatment-free survival time and no impact on overall survival (OS) was registered [20].

Management of early disease includes Binet stage A and B without active disease; Rai 0, I and II without active disease. Treatment for this phase is the “watch and wait” strategy. It is also recommended to perform blood counts and clinical examinations every 3–12 months. Pneumococcal vaccination as well as seasonal influenza vaccination is recommended in early-stage CLL [21]. Studies have not shown a beneficial effect of chemotherapy treatment in the early stages of the disease [22].

Advanced treatment strategy is initiated when the disease symptoms occur or the disease is diagnosed in an advanced clinical stage with bone marrow failure. Autologous stem-cell transplantation has no use in patients with CLL [23]. Allogeneic Stem Cell Transplantation (alloSCT) should be considered in patients who achieved remission taking kinase inhibitors or BCL2 antagonists after early relapse from chemoimmunotherapy and/or with del (17p) or TP53 mutation. The decision to start alloSCT therapy should be undertaken after careful consideration of the risks and benefits of allogeneic transplantation, considering, e.g., the patient’s age and comorbidities [24].

## 3. Materials and Methods

This systematic literature review was conformed in accordance with PRISMA Statement guidelines.

### 3.1. Objective

The purpose of the present study was to integrate the available data published in the literature on patients with chronic lymphocytic leukemia and to discuss and evaluate characteristics, insights into the pathogenesis of CLL and how humoral immunodeficiency influence clinical management and the development of therapeutic strategies for this disease.

### 3.2. Search Strategies

A systematic bibliographic search was performed in the PubMed database for all relevant studies published up to January 2020, investigating the association between chronic lymphocytic leukemia, secondary immunodeficiency and complying therapy. The following terms were used in the search strategy: (Chronic lymphocytic leukemia* OR CLL*) AND (targeted therapy* OR therapy* OR treatment*) AND (humoral immunodeficiency* OR secondary immunodeficiency* OR secondary immunodeficiency disorders*). The search strategy resulted in 240 articles. Additionally, Google Scholar was also used to evaluate search results and support manual research in journals related to immunology and hemato-oncology.

### 3.3. Selection Criteria

Two authors independently reviewed the results of the preliminary search. In case of duplicate reports or studies with clearly reported similar results, researchers chose the most recent and complete studies.

#### 3.3.1. Included Criteria

Studies evaluating the humoral immunodeficiency in chronic lymphocytic leukemia, pathogenesis or treatment;Articles published in the last twenty-five years with the latest information and novel guidelines.

#### 3.3.2. Exclusion Criteria

Restrictions on document type: technical report, editorial, letter, conference summary;Articles in a language other than English or Polish;The research contained incomplete content or full text of the article was not available;Studies that strictly evaluate only animal-related research.

## 4. Results

### Literature Search

The initial database search generated 240 articles, 158 of which were excluded because of duplicates. After reading the title and abstract, 158 records were excluded via the initial screening. The remaining 43 articles were collected and analyzed, excluding another 26, for the following reasons: type of article (one study); studies focused on strictly genetic correlation or only animal-related studies (four studies); publications in a different language (four studies); research published earlier than 1995 (ten studies); studies for which full text was not available (two studies). Finally, 22 studies met all the inclusion criteria and were included in the systematic review. The PRISMA flow diagram (Figure 1) illustrates the complete selection process.

## 5. Discussion

### 5.1. Secondary Immunodeficiency Disorders in Chronic Lymphocytic Leukemia

In CLL, deregulation of the immune system is a key feature from the early stages of the disease and worsens with clinical follow-up. T lymphocytes and immunoglobulins are elevated in early CLL and show progressive accumulation and depletion in advanced stages of disease. There are both quantitative and qualitative defects in immunity that result in abnormal cellular and humoral-mediated immune responses (Figure 2). One of the most frequently diagnosed immunological defects in patients with CLL is hipogammaglobulinemia, which mainly affects class IgG3 and IgG4 immunoglobulins. This condition is the result of a reduced number of normal B cells as well as a disruption of their regulatory cells. Moreover, an inadequate B cell response to IL-2 is also frequently observed, which also contributes to hipogammaglobulinemia. Complement levels are also decreased in CLL patients, in particular the C3b fraction, which increases the incidence of recurrent bacterial infections, but there are also defects in the activation, binding and expression of the CR1 (complement receptor 1) and CR2 (complement receptor 2) complement receptors. CLL patients also suffer from impaired cellular immunity. Often the number of helper T cells is decreased while the percentage of suppressor T cells is increased. Additionally, in CLL patients, CD3^+^CD8^+^ T cells produce a reduced amount of IL-2 as well as increased amounts of INF-γ and TNF-α [6,25,26,27]. Prompt diagnosis of secondary antibody deficiency is key to reduce the burden of infection and is dependent on appropriate screening and assessment of risk factors for developing secondary antibody deficiency.

Monitoring patients at risk of secondary antibody deficiency, such as those receiving conventional immunosuppressants or newer, more targeted therapies, can help identify these patients before they develop a serious infection [28]. This is especially important in the current COVID-19 (Coronavirus Disease 2019) pandemic situation. A case of CLL of a patient with secondary immunodeficiency who tested positive for SARS-Cov-2 has been reported [29]. After the patient’s condition worsened, he was also diagnosed with co-infection with parainfluenza 4 and further progression of CLL [29]. In this patient, an unconventional therapeutic approach was applied in the form of high-flow oxygen supplementation via nasal cannula, continuous monitoring of the patient’s condition to detect the possibility of co-infection and finally immunoglobulin substitution, which, as a result, contributed to the patient’s recovery [29]. All indications are that the current approach to treating patients with CLL who will develop COVID-19 should be changed, which may result in a greater number of convalescents in the future.

### 5.2. Humoral Immunodefficiency—Clinical Implications

Although the percentages rates in the total number of deaths in patients with CLL are diversified, infections are a major factor in the morbidity and mortality of CLL patients. Wadhwa et al. show that infections ultimately cause 30% of all CLL deaths [30], while other researchers present slightly different data, with Molica et al. [5] suggesting that they account for 50–60% of all deaths. Pathogenesis of infection in CLL is complex. Both the primary disease process and long-term immunosuppressive therapy is significant. The 5-year risk for severe infections in CLL is 26% overall, and increases to 57% once IgG levels are low, and up to 68% for patients with both low IgG levels and Binet stage C [31].

Infections remain a major contributor to morbidity and mortality in patients with CLL, ultimately contributing high percentages of CLL deaths [30].

However, it seems that hipogammaglobulinemia is the key cause of the increased incidence of infectious complications. Successful treatment and complete remission of CLL do not usually affect the Ig-count. Nonetheless, a recent study shows that high doses of rituximab can partially restore antibody production [32]. The relationship between the level of a specific class of antibody and the type of pathogen causing the infection is controversial among researchers. However, most agree that low IgG levels are associated with recurrent infections with *Streptococcus (S.)* and *Haemophilus (H.)* [27]. Recently, the influence of the deficiency of immunoglobulins on the mucous membranes has been investigated. The level of IgM antibodies in saliva of patients was shown to be diminished with a low serum IgA level [6]. Predominant IgA deficiency is associated with an increased incidence of upper respiratory tract infections. The role of the deficiency of antibodies in mucous membranes has not been fully investigated and further research is needed. From another point of view, not all patients with CLL and hipogammaglobulinemia suffer recurrent infections. Researchers agree that there is no reliable method for determining the qualitative function of immunoglobulins to associate it with recurrent infections in CLL. Moreover, the risk of recurrent and severe infections increases with the duration and the stage of the disease. Infectious episodes are more severe and more frequent in patients with Binet stage C (82%) than in patients with stage A (33%). Chemoimmunotherapy, anti-CD-20 (rituximab, ofatumumab, obinutuzumab) and anti-CD-52 (alemtuzumab) antibodies as a modern treatment method has almost completely replaced purine analogues and alkylating agents used alone. Recently, tyrosine kinase inhibitors (ibrutinib and idelalisib) and Bcl2 antagonists (ABT-199) have been introduced. The studies of Severin et al. reported that the use of ibrutnib in combination with JAK2 and STAT3 inhibitors significantly increases the tumor cell death induced by ibrutinib, even in the presence of bone marrow mesenchymal stromal cells (BMSCs), which protect tumor cells from removal [33]. Treatment initiation with alkylating agents is likely to induce myelosuppression, which also increases the risk of infections. Epidemiological factors of infections associated with the intake of alkylating agents are *S. aureus*, *S. pneumoniae*, *H. influenzae*, and *Klebsiella pneumoniae*. Further the use of purine analogues affects DNA synthesis and impairs the number of B cells and monocytes. Therefore, opportunistic infections caused by *Listeria monocytogenes*, *Mycobacterium* spp., *Pneumocystis jirovecii*, *herpes simplex virus*, *varicella-zoster virus*, *Candida* spp. are common [2,34]. The CLL-4 trial showed that the alkylating agents combined with purine analogs in multi-drug therapy can cause thrombocytopenia and leukocytopenia. The study also showed that the number of serious infections did not increase compared to fludarabine monotherapy [35]. Alemtuzumab therapy requires monitoring of CMV once a week, and in the case of CMV antigenemia, antiviral treatment is obligatory (ganciclovir) [36].

National Comprehensive Cancer Network guidelines recommend that patients with CLL and symptomatic Cytomegalovirus (CMV) infection or CMV reactivation should be hospitalized and treated with ganciclovir or valganciclovir for at least 2 weeks. General antifungal prophylaxis is not recommended. The analysis of data from 795 people with CLL showed that the number of previous chemotherapy treatments and the level of immunoglobulins were important for fungal infections [37]. Receiving purine analogues or alemtuzumab is also associated with an increased likelihood of fungal infections. The most frequently administered antifungal drug is fluconazole. *Aspergillus* spp. infection suspicion obliges to administer itraconazole, voriconazole, posaconazole or caspofungin [38,39].

### 5.3. Hummoral Immunodefficiency—Pathogenesis

Many pathological conditions underlie cellular changes resulting from normal physiological mechanisms operating outside their proper context. This is especially important for the molecular interactions that control cellular apoptosis given the deleterious effects of such mechanisms in the absence of strict control. Publications provide evidence explaining the pathogenesis of humoral immunodeficiency, linking this molecular relationship.

A long duration of CLL and its severity correlates with decreased levels of serum IgG, IgA, and IgM antibodies [40]. Moreover, the decrease in the concentration of antibodies of all classes occurs in most patients independently of mutations in the immunoglobulin heavy chain genes (IGHV) and treatment stage [41].

Initially, researchers postulated that the cause of hipogammaglobulinemia was a defect in cells that regulate the maturation of normal B cells, excessive suppression of lymphocytes T, decreased T helper function, abnormal response to IL-2 or massive accumulation of leukemic B lymphocytes diluting normal B lymphocytes [42,43,44]. Tinhoffer et al. [45] proved the presence CD95L on the membrane of leukemic cells, with the molecule being a natural ligand for CD95 (death receptor, Fas receptor, APO-1). Further, increased expression of surface CD95 on patients’ CD4^+^ T cells was proven. These observations led the authors to hypothesize that B-CLL cells could directly eliminate the helper T cell function through this mechanism, which would, in turn, cause the humoral immunodeficiency in CLL. The authors concluded that B-CLL cells could directly affect the T helpers’ function and number, causing humoral immunodeficiency [45,46].

The results of these studies prompted subsequent scientists to investigate the mediated inhibitory mechanism of B-CLL cells on plasmatic cells (PC). Plasma cells are found in the bone marrow and their assignment is to produce a large quantity of proteins—antibodies (immunoglobulins)—in response to being presented with specific substances called antigens. Sampalo et al. [47] revealed the inhibitory effect of B-CLL cells on the plasmatic cells. The inhibitory effect on the production of IgM, IgG, and IgA antibodies by PC was proportional to the increase in the number of B-CLL cells. Moreover, direct contact between B-CLL cells and plasma cells was necessary to obtain the inhibitory effect. Subsequent experiments showed the presence of CD95 on PC (and lack of CD95L). In contrast, B-CLL cells did not show the presence of CD95, although previously mentioned CD95L was present on their surface. It can, therefore, be concluded that B-CLL cells inhibit autologous antibody production by inducing apoptosis of plasma cells through the CD95L–CD95 interaction. This may result in humoral immunodeficiency in these patients [44,47]. The studies by Cerruti at al. [48] and Cantwell at al. [49] prove the influence of B-CLL cells on regulatory T lymphocytes, the proper function of which is essential for Ig class-switching in the early stages of differentiation.

### 5.4. Treatment

As stated above, only treating the CLL does not restore immunity and guidelines do not recommend immunodeficiency as a reason for starting treatment. In hematological malignancies, supportive treatment, including prophylactic vaccination (with non-live vaccines), antibiotics and/or IgRT in patients with secondary antibody deficiency, may be considered. A careful review of the patient’s history, assessment of risk factors for developing secondary antibody deficiency, and assessment of serum IgG levels and specific antibody levels is key to diagnosis and treatment [28].

#### 5.4.1. Prophylactic Antibiotics

Antibiotic prophylaxis is the recommended first-line treatment for symptomatic antibody deficiency in CLL and during periods of neutropenia in patients undergoing chemotherapy or other immunosuppressive therapies [50,51]. The choice of antibiotic will largely depend on the patient’s history of infection, therefore, previous culture and sensitivity should be considered, as well as any allergies and tolerance and probability of infection with macrolide-resistant *H. Influenzae* or *Pseudomonas aeruginosa* should be considered. In the event of a breakthrough infection, intravenous antibiotic treatment (IVAB) should be considered, once the response to an additional course of antibiotics and a second course of other antibiotics has not been achieved or has been limited and not fully effective. It is worth noting that prophylactic and supplementary antibiotics should belong to different classes (e.g., macrolides and penicillin) and not simply increase the dose within the existing prophylactic regimen. Moreover, with long-term use of macrolides, the patient should be given additional information about side effects (tinnitus) and monitoring (electrocardiogram). There are many potential antibiotic options, and examples of individual decisions are made based on clinical and local prescribing policies. Data presented in the literature show that azitomycin is a frequently used antibiotic, is effective against common respiratory pathogens and has good penetration through the mucous membranes, a long half-life allowing dosing approximately three times a week and established efficacy in the treatment of bronchiectasis. In contrast, nebulized antibiotics and intermittent IVAB are options primarily used for severe bronchiectasis and pseudo-colonization [51].

#### 5.4.2. The Use of Immunoglobulin Replacement Therapy

Intravenous immunoglobulin (IVIG) is prepared from pools of plasma obtained from several thousand healthy blood donors. The large donor pool provides a variety of antibody repertoires that encompass the specificity of the antibodies for a broad spectrum of antigens [52,53]. IVIG contains a sample from a complete set of antibody variable regions similar to those present in normal human serum. Antibodies that arise in the absence of pathological conditions or deliberate immunization are called natural antibodies (NAb), and IVIG is a privileged source of natural antibodies. NAb has been assigned a variety of functions, including the first line of defense against pathogens. The presence of antibodies to recurring pathogens may therefore be of key importance in the replacement therapy of patients with humoral immunodeficiency [54,55].

Many studies have demonstrated decreased infection incidence in patients with CLL treated with intravenous IgG. A study by Boughton et al. [56] describes forty-two patients with chronic lymphocytic leukemia (CLL), serum IgG levels < 5.5 g/L and a history of two or more recent infections, were randomized to receive infusions of 18 g human intravenous immunoglobulin (IVIg) or human albumin placebo every three weeks. During the 12-month study, 122 infections were documented but only four were associated with neutropenia. Ten patients (24%) with IgG levels < 3.0 g/L experienced 65% of the infections. In response to IVIg there were immediate and accumulative increases in serum IgG levels and an associated decrease in total and serious infections. If three further infections occurred, placebo patients were commenced on 18 g IVIg, and IVIg patients were increased to 24 g IVIg. Approximately 50% of these cases subsequently remained infection-free. The study demonstrates the usefulness of prophylactic immunoglobulin in CLL patients with hipogammaglobulinemia and suggests that it may be justified in patients with recurrent infections and serum IgG < 3 g/L [56].

A trial conducted by Molica et al. [57] included 42 CLL patients with hipogammaglobulinemia (IgG < 600 mg/dL) and/or a history of at least one episode of severe infection in the 6 months preceding inclusion. Patients were randomly allocated to receive either an infusion of 300 mg/kg IVIG every 4 weeks for 6 months or no treatment. Afterwards, they were switched to observation or IVIG for another 12 months; finally, they received IVIG or no therapy for 6 more months. A significantly lower incidence of infectious episodes was observed during IVIG prophylaxis in 30 patients who completed the 6-month period of either observation or IVIG therapy. The same applied to the 17 patients who completed 12 months of either observation or IVIG prophylaxis. Interestingly, the restoration of serum IgG levels obtained in 17 out of 25 patients (mean percent value of IgG increase, 41.8%) did not parallel a decrease in the number of infectious episodes. A protective effect against infections is demonstrated for low-dose IVIG in the present study. A benefit was shown in patients who completed either 12 or 6 months of IVIG prophylaxis; however, even this low-dose treatment is not a cost-effective way to prevent infection in CLL patients [57]. However, since these studies are more than twenty years old, there is a need for more recent randomized controlled trials in this area, especially given the tremendous advances in treatment and increased survival. The IgRT protocols for secondary antibody deficiency differ. The key factors to consider when initiating IgRT therapy is the selection of those who may benefit. Studies investigate whether patients with IgG levels < 4 g/L and/or low levels of antibodies against encapsulated organisms with an ongoing history of recurrent bacterial infections that have failed to respond to prophylactic antibiotics may benefit from additional IVIG. It has also been suggested that the use of IgRT in patients who have previously received more than three treatment lines, along with prophylactic use of antifungal medications, may be effective in reducing the risk of fungal infections [37]. The benefit/risk ratio should be carefully assessed during treatment with IgRT. Important factors in the selection of patients are the assessment of comorbidities, immune disorders—neutropenia and demonstration of antibody failure (exposure/test vaccination). In addition, IgRT side effects such as thromboembolic complications and hemolysis in patients with hematological malignancies should be considered [56,58]. In addition, two methods can be given for Ig therapy; intravenous (IVIG) and subcutaneous (SCIG) also consider the location (home vs. clinic) of the infusion. Venous access is often a serious problem after chemotherapy and SCIG avoids the use of venous accesses. The flexibility of SCIG treatment and the possibility of home infusion is another step forward for patients who usually require a large number of outpatient visits, such a solution significantly reduces costs and improves the patient’s quality of life [59]. According to the guidelines of the European Medicines Agency (EMA) 2018 IVIG can be used in patients with secondary immunodeficiencies “who suffer from severe or recurrent infections, ineffective antimicrobial treatment and either proven specific antibody failure (PSAF) or serum IgG level of <4 g/L,” where PSAF is defined as “failure to mount at least a 2-fold rise in IgG antibody titer to pneumococcal polysaccharide and polypeptide antigen vaccines”. The required dose is said to be possibly patient-dependent, but is likely to be in the range of 0.2–0.4 g/kg every 3–4 weeks [60].

With regard to subcutaneous use (SCIG), EMA guidelines suggest replacement therapy in “Hipogammaglobulinemia and recurrent bacterial infections in patients with chronic lymphocytic leukemia (CLL), in whom prophylactic antibiotics have failed or are contra-indicated”. However, the EMA states that “the above indications would be granted as long as efficacy has been proven in primary immunodeficiency syndromes” [61]. The European Society for Medical Oncology (ESMO) guidelines state that “the use of prophylactic systemic IgG (in patients with CLL) does not have an impact on overall survival and is only recommended in patients with severe hipogammaglobulinemia and repeated infections.” They also state that “Antibiotic and antiviral prophylaxis should be used in patients with recurrent infections and/or very high risk of developing infections (e.g., pneumocystis prophylaxis with co-trimoxazole during treatment with chemoimmunotherapies based on purine analogs or bendamustine)” and finally, that “pneumococcal vaccination as well as seasonal influenza vaccination is recommended in early-stage CLL” [62].

Overall, most guidelines support considering initiation of IgRT therapy in selected patients with secondary antibody deficiency. However, further clinical trials are needed to validate the standardization and refinement of the current practice of prescribing IgG to patients with secondary immunodeficiency in CLL, especially as new therapies develop.

## 6. Conclusions

The clinical heterogeneity of CLL patients seems to reflect the diversity of the molecular abnormalities that drive the pathogenesis and progression of disease. Through a better understanding of the molecular genetics and biology of the CLL cell, new biomarkers have been invented to stratify and determinate targets for the development of novel and more selective treatments. Nevertheless, it is infections that remain the leading cause of morbidity and mortality in CLL. Humoral immunodeficiency in CLL is associated with an increased risk of infection. Early diagnosis and intervention are critical for proper treatment. Optimizing treatment requires careful clinical and laboratory evaluation and may involve close monitoring of risk parameters, vaccination, antibiotic strategies, and in some patients, immunoglobulin replacement therapy (IgRT).

## Figures and Tables

**Figure 1 cells-09-02398-f001:**
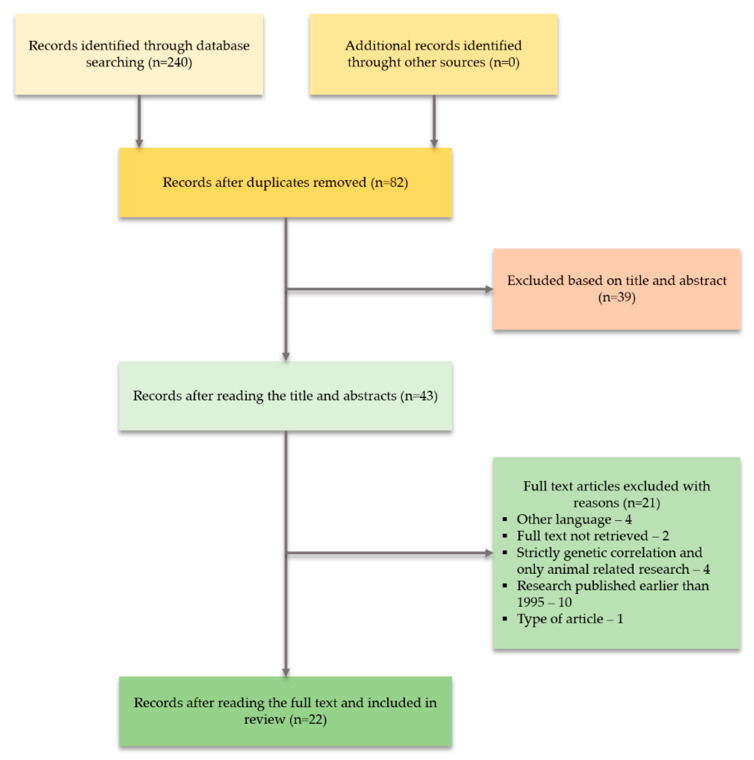
PRISMA flow diagram.

**Figure 2 cells-09-02398-f002:**
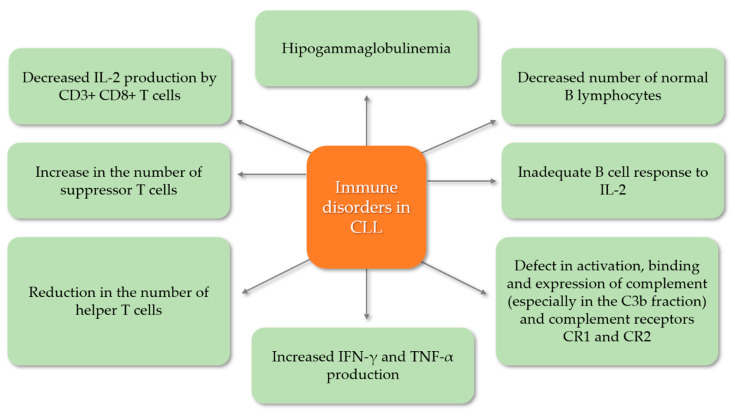
Immune defects in chronic lymphocytic leukemia.

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
