# Peer review of "Chronic Lymphocytic Leukemia-Induced Humoral Immunosuppression: A Systematic Review"

_cells, 2020, doi:10.3390/cells9112398_

Round 1
Reviewer 1 Report
The aim of this review is to present the already published works on CLL with regard to the pathogenesis of the disease and the impact of humoral immunity deficiency on the clinical and therapeutic approach. The authors performed a literature research using appropriate filters and focused the analysis on 22 studies.
Overall, this review is well structured. The authors structured the review in 5 main part. I appreciated in the mat&Met the explanation of the research methods and the figures help the reader. However, I have some important suggestions:
- to revise the entire manuscript for the English, I found several grammatical and typing errors. For instance, in line 142-143 “Thus, increased expression of BCR genes is present in 142 all patients with CLL in lymph nodes and lymph nodes”, probably one “lymph nodes” have to substitute by “bone marrow”?
- I suggest adding a section or considerations regarding to the course of COVID-19 in a CLL patient with secondary immunodeficiency and viral co-infection with parainfluenza. This could be given a potential improvement of the manuscript and helped the knowledges and the future approaches for CLL treatment.
In my opinion it is necessary a revision of this manuscript to make it more appealing. It is interesting the topics of this review, but I suggest performing the analysis also using “Covid-19” as a filter.
Author Response
Dear Reviewer1,
On behalf of the authors of the review article entitled "Chronic lymphocytic leukemia-induced humoral immunosuppression: a systematic review" by Grywalska, E.; Zaborek, M.; Łyczba, J.; Hrynkiewicz, R.; Bebnowska, D.; Becht, R.; Sosnowska-Pasiarska, B.; Smok-Kalwat, J.; Pasiarski, M.; Góźdź, S.; Roliński, J.; Niedźwiedzka-Rystwej, P., we would like to cordially thank you for your review. We tried to follow all your suggestions and corrected the manuscript accordingly. Here are the point-by-point answers, all changes to the manuscript are marked in red:
- to revise the entire manuscript for the English, I found several grammatical and typing errors. For instance, in line 142-143 “Thus, increased expression of BCR genes is present in 142 all patients with CLL in lymph nodes and lymph nodes”, probably one “lymph nodes” have to substitute by “bone marrow”?
- Thank you for your valuable attention. As suggested by the reviewer, we decided to review the entire text to rule out more grammatical and spelling errors. Additionally, we improved lines 142-143 (now 108-109). We changed a word "lymph nodes" on the word "bone marrow"? We apologize for the mistake.
- I suggest adding a section or considerations regarding to the course of COVID-19 in a CLL patient with secondary immunodeficiency and viral co-infection with parainfluenza. This could be given a potential improvement of the manuscript and helped the knowledges and the future approaches for CLL treatment.
- As suggested by the reviewer, we added a paragraph (lines 207-216) about the course of COVID-19 in a CLL patient with secondary immunodeficiency and parainfluenza viral co-infection.
Again, we would like to thank you for your effort and time and we are hoping that our manuscript in its current form will fulfill the requirements of the Cells.
Thank you for your time and consideration,
Paulina Niedźwiedzka-Rystwej
Reviewer 2 Report
This study is a systematic Review of the humoral immunosuppression in chronic lymphocytic leukemia (CLL). It describes the clinical aspects, pathogenesis and treatment.
It is clear and well written, however, the review is too long and it is not focus on the key aspects of the topic. For instance, the introduction is a good revision of this disease for a book, but, from my point of view, it is not adequate for a review. It is too long and dedicate a lot of paragraphs for information that is not relevant for this topic. For instance, Table 1 is irrelevant for this review.
Contrarily, the review does not give any information about the immune aspects of the CLL that are more relevant for this topic. The figure 2 provides just a little information about the immune response in CLL that it is quite interesting, but the information provided in the figure is not explained anywhere and it is not cited where this information comes from. Overall, the introduction should be shorter and focus on key aspects related to this topic.
It is unusual to do a systematic review for this kind of topics. The search criteria is quite limited and some important information about immunodeficiency in CLL may be missed.
The figure 1 is irrelevant. It should be eliminated.
Author Response
Dear Reviewer2,
On behalf of the authors of the review article entitled "Chronic lymphocytic leukemia-induced humoral immunosuppression: a systematic review" by Grywalska, E.; Zaborek, M.; Łyczba, J.; Hrynkiewicz, R.; Bebnowska, D.; Becht, R.; Sosnowska-Pasiarska, B.; Smok-Kalwat, J.; Pasiarski, M.; Góźdź, S.; Roliński, J.; Niedźwiedzka-Rystwej, P., we would like to cordially thank you for your review. We tried to follow all your suggestions and corrected the manuscript accordingly. Here are the point-by-point answers, all changes to the manuscript are marked in red:
- It is clear and well written; however, the review is too long and it is not focus on the key aspects of the topic. For instance, the introduction is a good revision of this disease for a book, but, from my point of view, it is not adequate for a review. It is too long and dedicate a lot of paragraphs for information that is not relevant for this topic. For instance, Table 1 is irrelevant for this review.
- Thank you very much for your valuable attention. Following the Reviewer's recommendations, we decided to significantly shorten the introduction. We removed two paragraphs describing CLLs and Table 1 and Table 2. Additionally, we decided to change Subchapter 1.1 to Chapter 2.
- Contrarily, the review does not give any information about the immune aspects of the CLL that are more relevant for this topic. The figure 2 provides just a little information about the immune response in CLL that it is quite interesting, but the information provided in the figure is not explained anywhere and it is not cited where this information comes from. Overall, the introduction should be shorter and focus on key aspects related to this topic.
- In line with the Reviewer's recommendations, we decided to add a paragraph describing the immune response in CLL (Figure 2) (Lines 193-203).
- It is unusual to do a systematic review for this kind of topics. The search criteria is quite limited and some important information about immunodeficiency in CLL may be missed.
- We understand the doubts of the Reviewer, that our systematic review may be unusual, but we hope that this unusual subject will meet the expectations of the Reviewers, Editors and Readers.
- The figure 1 is irrelevant. It should be eliminated.
- After analyzing the Reviewer's remark, we decided to leave Figure 1, because the diagram presented by us is characteristic for systematic review.
Again, we would like to thank you for your effort and time and we are hoping that our manuscript in its current form will fulfill the requirements of the Cells.
Thank you for your time and consideration,
Paulina Niedźwiedzka-Rystwej
Reviewer 3 Report
The manuscript is fine. However, there are a lot of errors in spelling and paragraph segmentation. For example, line 228 and Figure 2, “hypogammaglobulinemia” should be “hipogammaglobulinemia”. In Figure 2, what is the difference between the upper figure and the lower figure? They are the same except colour. The manuscript should be carefully and extensively revised before submission.
Author Response
Dear Reviewer,
Thank you for your kind consideration of our paper. We are grateful for the mistakes that you have noticed and we did our best to correct them to improve the quality of the manuscript.
All (noticed by us) errors in spelling and paragraph segmentation were corrected, with a special attention to “hipogammaglobulinemia”. The Figure 2 has been changed – colors were erased, as they were misleading, and also two “windows” of the Figure expressing very similar concepts were combined into one.
Also, we read the manuscript again, to avoid any other mistakes.
Again, we would like to thank you for your effort and time and we are hoping that our manuscript in its current for will fulfill the requirements.
Best regards,
Round 2
Reviewer 1 Report
I revised the manuscript after authors response report, and I appreciated the revision of entire manuscript for English and the addition of my request.
Author Response
Dear Reviewer,
Thank you very much for you time and consideration of our manuscript.
Best regards,
Reviewer 2 Report
From my point of view, this manuscript is not adequate for publication. I do not want to review it again.
Author Response
Dear Reviewer,
We understand your concerns. We have already known your opinion since your first review. Nevertheless, we would like to try our best to improve, that is why we asked the Editors for an additional round of reviewing to make the process as objective as possible. Thank you again for your time and consideration.
Best regards,